# GENN: Predicting Correlated Drug-drug Interactions with Graph Energy Neural Networks

## Abstract

Gaining more comprehensive knowledge about drug-drug interactions (DDIs) is one of the most important tasks in drug development and medical practice. Recently graph neural networks have achieved great success in this task by modeling drugs as nodes and drug-drug interactions as links and casting DDI predictions as link prediction problems. However, correlations between link labels (e.g., DDI types) were rarely considered in existing works. We propose the graph energy neural network (GENN) to explicitly model link type correlations. We formulate the DDI prediction task as a structure prediction problem, and introduce a new energy-based model where the energy function is defined by graph neural networks. Experiments on two real world DDI datasets demonstrated that GENN is superior to many baselines without consideration of link type correlations and achieved 13.77% and 5.01% PR-AUC improvement on the two datasets, respectively. We also present a case study in which GENN can better capture meaningful DDI correlations compared with baseline models.

## 1 Introduction

The use of drug combinations is common and often necessary for treating patients with complex diseases. However, it also increases the risk of drug-drug interactions (DDI). DDIs are pharmacological interactions between drugs that can alter the action of either or both drugs and cause adverse effects. Overall DDIs result in a large number of fatalities per year and incur $\sim$ \$177 billion DDI associated cost annually (Giacomini et al., 2007). To mitigate these risks and costs, accurate prediction of DDI becomes a clinically important task.

While DDI knowledge is expensive to gather, several deep learning approaches have been proposed to search the large biomedical data for predicting potential DDIs (Zitnik et al., 2018; Ma et al., 2018; Ryu et al., 2018). Among them, graph neural networks (GNN) that consider DDI prediction in the graph setting have obtained great performance. In DDI graphs, drugs are represented as nodes and embedded into a low-dimensional space. Further prediction can be made based on casting DDI prediction as a link prediction problem where DDI types are represented as link types (Kipf & Welling, 2016; Zitnik et al., 2018).

However, existing works rarely exploit the correlations between these link types (e.g., DDI types) despite of their importance. For example, in Fig. 1 Warfarin is a drug for treating blood clots. Its combined use with antibiotic or nonsteroidal anti-inflammatory drugs can cause multiple DDI types including nhibition of clotting, gastrointestinal bleeding, and hemorrhage. These DDI types are correlated since they are all bleeding related DDIs caused by the increase of Warfarin's effect. Explicit modeling such correlations can help infer unseen DDI types.

To fill the gap, we propose GENN, a new deep architecture that predicts correlated DDIs based on graph neural networks and energy based models. Specifically, we leverage the dependency structures among DDI types and formulate multi-type DDI detection as a structure prediction problem (Lafferty et al., 2001a; LeCun et al., 2006; Belanger & McCallum, 2016b). Here we use an energy based approach to incorporate such dependency structures.

To summarize, GENN is enabled by the following technical contributions.

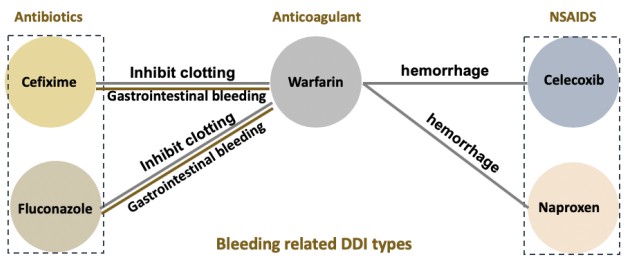

Figure 1: Warfarin as an anticoagulant, if taken together with antibiotic drugs such as Cefixime or Fluconazole, will have high risk to cause DDIs **inhibition of clotting** and **gastrointestinal bleeding**. Likewise, if Warfarin if concomitant with nonsteroidal anti-inflammatory drugs (NSAIDs) such as Celecoxib and Naproxen, it will likely to cause DDI **hemorrhage** (i.e., loss of blood from broken blood vessel). These DDIs are correlated since they are all bleeding related DDIs caused by the increase of Warfarin's effect.

1. **Modeling link type correlations**. GENN bridges graph neural networks and structure prediction to directly capture link type correlations for more accurate link prediction in graphs through minimizing an energy function.
2. **A new graph neural network based energy function**. Inspired by a family of structure prediction model called structured prediction energy networks (SPENs) (Belanger & McCallum, 2016a), we design a new graph energy function based on graph neural networks to capture the global dependencies among link types in DDI graphs. This is more flexible and expressive than previous approaches which used CRF-type energy definitions Qu et al. (2019).
3. **Efficient semi-supervised training**. We designed one cost-augmented inference network to approximate the output in training and one test inference network to approximate the output in testing. Different from the previous inference method for SPENs, we propose a new semi-supervised joint optimization procedure to adapt to our problem on graphs. The new inference method optimizes the training-set structured hinge loss together with the test-set energy with respect to the parameters for both inference networks and the energy function.

We evaluated GENN on two real world DDI datasets with both quantitative and qualitative study. Results demonstrated GENN outperformed basic graph neural networks with 13.77% and 5.01% PR-AUC improvement on the two datasets, respectively.

## 2 RELATED WORK

### 2.1 DDI PREDICTION

To predict unseen DDIs based on known ones, drug similarity has been learned via nearest neighbor approaches (Zhang et al., 2017), and using random walk methods including label propagation (Zhang et al., 2015; Wang et al., 2010), and unsupervised methods (Wang et al., 2014; Angione et al., 2016). Recently, deep graph neural networks have been demonstrated to provide much improved performance in DDI prediction. Among them, (Ma et al., 2018) integrates different sources of drug-related information with heterogeneous formats into an coherent and information-preserving representation using attentive multi-view graph auto-encoders (Kipf & Welling, 2016). Decagon (Zitnik et al., 2018) develops a new graph auto-encoder approach, which allows us to develop an end-to-end trainable model for link prediction on a multi-modal graph. As the experimental results in Zitnik et al. (2018) show graph neural networks achieved significantly improved performance than shallow models such as tensor decomposition (Nickel et al., 2011; Papalexakis et al., 2017), random walk based methods (Perozzi et al., 2014; Zong et al., 2017), and non-graph neural fingerprinting (Duvenaud et al., 2015; Jaeger et al., 2018), DeepDDI (Ryu et al., 2018). None of the existing DDI prediction works consider the correlation among multiple DDI types.

## 2.2 STRUCTURE PREDICTION AND ENERGY BASED MODELS

Structure prediction is an important problem in various application domains where we want to predict structured outputs instead of independent labels, e.g., structured labels prediction for object detection (Zheng et al., 2015b), semantic structure prediction (Belanger et al., 2017) or part-of-speeches tagging (Ma & Hovy, 2016). For structure prediction problems, feed-forward networks is insufficient since they cannot directly model the interaction or constraints of the outputs. Instead, energy based models (LeCun et al., 2006) provide one way to address the challenge by defining a much more flexible and expressive energy score, and the prediction is conducted by minimizing the energy function.

One of the well-known structure prediction approaches is the conditional random fields (CRF) (Lafferty et al., 2001b) which show success in application areas such as named entity recognition (Sato et al., 2017) and image segmentation (Zheng et al., 2015a). But as a structured linear model, CRF has limited representation ability. To enhance the flexibility of structure prediction models, Belanger & McCallum (2016b) proposed *Structured Prediction Energy Networks (SPENs)* which uses arbitrary neural networks to define the energy function and optimizes the energy over a continuous relaxation of labels. Thus it can capture high-arity interaction among output labels and approximate the optimization of energy function via gradient descent and repeated "loss-augmented inference". Belanger et al. (2017) further developed an "end-to-end" method that unroll the approximate energy optimization to a different computation graph. However, after learning the energy function, they still have to use gradient descent for test-time inference. Later, Tu & Gimpel (2018) replaced the gradient descent with a neural network trained to do inference directly. However, it separates the training of cost-augmented inference network and the fine-tuning of another inference network for testing, which makes it inefficient. In addition, how to optimize the SPENs on graphs for semi-supervised learning remains challenging. In this paper our method improves the algorithm of Tu & Gimpel (2018) to solve this problem.

Very recently, GMNN (Qu et al., 2019) combines the statistical relational learning (SRL) and graph neural networks. It includes a CRF to model the joint distribution of object labels conditioned on object attributes, and then it is trained with a variational EM algorithm. The model has been used for node classification and link classification. However, it cannot be easily extended for prediction of missing links, since they have to use links as nodes in a "dual graph" for link classification, but the missing links generally cannot be modeled as nodes.

## 3 PRELIMINARY

In this section, we first summarize our task in Section 3.1, then a basic message passing graph neural network (MPNN) is described for embedding the DDI graph in Section 3.2.

### 3.1 TASK FORMULATION

**Definition 1 (DDI Graph)** *Given a DDI graph $G = \{\mathcal{V}, \mathcal{E}\}$, $\mathcal{V}$ is the node set which contains $N$ drug nodes with node features $\mathbf{X} \in \mathbb{R}^{N \times D}$, and $\mathcal{E}$ is the edge set which contains all DDIs between drug pairs. For a specific drug pair $\mathbf{x}_i \in \mathbb{R}^D, \mathbf{x}_j \in \mathbb{R}^D$, the DDIs of this pair $\mathbf{e}_{i,j}$ can have multiple types, i.e. $\mathbf{e}_{i,j} \in \{0,1\}^L$, where $D$ is the feature dimension and $L$ is the total number of DDI types. And these DDI vectors $\mathbf{e}$ form the edge features (or labels) $\mathbf{Y} \in \{0,1\}^{|\mathcal{E}| \times L}$.*

**Problem 1 (DDI Prediction)** *We cast the DDI prediction into a multi-type link prediction problem. We assume there are some missing edges $\mathbf{Y}_U$ in the graph, i.e. $\mathbf{Y} = \{\mathbf{Y}_L, \mathbf{Y}_U\}$. Given the node features $\mathbf{X}$ and some known DDI links $\mathbf{Y}_L$, the goal is to predict the unknown DDIs $\mathbf{Y}_U$.*

### 3.2 MESSAGE PASSING NEURAL NETWORKS FOR DDI PREDICTION

Recently graph neural networks (GNNs) have been successfully applied to DDI prediction (Zitnik et al., 2018; Ma et al., 2018). To predict the unknown DDIs in a graph, a GNN first learns the node embeddings of all the drug nodes and then make prediction based on these embedding. In this

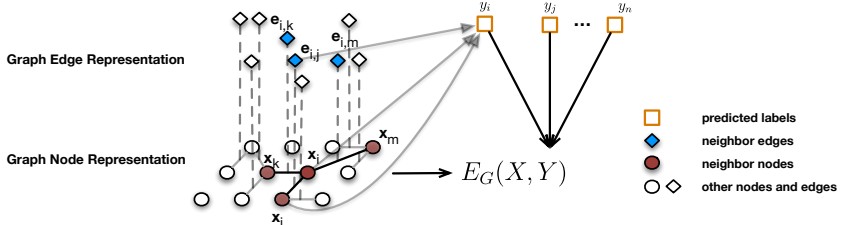

Figure 2: Graphical structure of GENN. Message passing procedure is conducted using Eq. 1 to update $i$-th node hidden state by aggregating neighbor edges and nodes information. Prediction is made for edge $e_{i,j}$ based on Eq. 2. Furthermore, we use an energy function via GNN to build interaction among all labels $Y$ globally as introduced in Eq. 3.

section, we describe a basic message passing based graph neural network (Gilmer et al., 2017) for drug node embedding. Considering that our graph contains different types of edges, we select a schema including edge information in the message passing. Given the graph $G$ with node features $\mathbf{X}$, for each node with hidden state $\mathbf{h}_i^t$ and its neighborhood $\mathcal{N}(i)$, we define the message passing layer to update the node hidden state as follows:

$$\mathbf{h}_i^{t+1} = \mathbf{W}_s \mathbf{h}_i^t + \sum_{j \in \mathcal{N}(i)} \mathbf{h}_j^t \cdot f(\mathbf{e}_{i,j}); \ \forall i \in \mathcal{V}, \mathbf{h}_i^0 = \mathbf{W}_0 \mathbf{x}_i \qquad (1)$$

where $\mathbf{e}_{i,j} \in \mathbb{R}^L$ is the edge (DDI type vector) between nodes $\mathbf{x}_i, \mathbf{x}_j \in \mathbb{R}^D$, $\mathbf{W}_s \in \mathbb{R}^{M \times M}$, $\mathbf{W}_0 \in \mathbb{R}^{M \times D}$ are both learnable weight, and $f(\cdot) \in \mathbb{R}^{M \times M}$ is a neural network, i.e. MLP.

The message passing layer can then be stacked to get the final state $\mathbf{h}_i$ for each node, i.e. the node embedding. Then we can use these final states to predict the probability of DDI types for each edge:

$$\hat{\mathbf{y}}_{i,j} = \text{Sigmoid}(\mathbf{W}[\mathbf{h}_i || \mathbf{h}_j] + \mathbf{b}) \qquad (2)$$

where $||$ indicates concatenation, $\mathbf{W} \in \mathbb{R}^{L \times 2M}$ and $\mathbf{b} \in \mathbb{R}^{2M}$ are parameters and Sigmoid is the activation function commonly used for multi-label prediction.

The model is trained on the partially known DDI edges $\mathbf{Y}_L$ by optimizing cross entropy loss $\mathcal{L}_{CE}(\hat{\mathbf{Y}}_L, \mathbf{Y}_L) = \sum_{\mathbf{e}_{i,j} \in \mathbf{Y}_L} \mathcal{L}_{CE}(\hat{\mathbf{y}}_{i,j}, \mathbf{e}_{i,j})$ between the predictions and the true edges. After training, it can be directly used for prediction of unknown DDIs $\mathbf{Y}_U$.

## 4 THE GENN METHOD

We present our GENN framework in this section (Fig. 2). We first discuss the design of energy based framework for DDI prediction and the design of our graph energy functions based on MPNN. Then we propose a joint learning strategy with a semi-supervised learning setting in Section 4.3.

Obviously the aforementioned message passing neural network does not explicitly consider the DDI correlation in the prediction. However, in practice, the labels are often correlated as we stated in the introduction section. So we re-formulate our problem as a structure prediction problem and infer the labels by optimizing an energy function. In order to get extremely powerful representation for the energy model, we follow the ideas of SPENs to formulate the energy function as a neural network.

### 4.1 ENERGY FUNCTION DESIGN

The SPENs define the energy $E(X, Y)$ as a neural network that takes both the input features $X$ and labels $Y$ as inputs and returns the energy. Although the SPENs can almost obtain arbitrarily high expressiveness, in practice it requires a trade-off between using increasingly expressive energy networks and being more vulnerable to overfitting (Belanger & McCallum, 2016b), so the energy function need to be properly designed.

**Energy function via GNN** Recall the message passing neural network we used before, the edge information is also included in the neural networks. If we aggregate all the nodes' and edges'

information to get the whole graph's representation, it is natural to derive an "energy" over the graph. Motivated by this intuition, we formulate a new and simple energy function from a graph neural network:

$$E(\mathbf{X}, \mathbf{Y}) = \text{Readout}(G_0(\mathbf{X}, \mathbf{Y})) = \text{Relu}(\text{MLP}(\frac{1}{N} \sum_i \mathbf{h}_i)) \tag{3}$$

Where $G_0$ is an arbitrary graph neural network (GNN) which accepts node features $\mathbf{X}$ and edge features $\mathbf{Y}$ which can be initialized by multi-hot encoding storing labels, $\mathbf{h}_i$ is the node embedding of final output of GNN which contains information of neighbor nodes and edges, MLP is a multi-layer perceptron network. Obviously, Eq. (3) contains both the interaction between each $\mathbf{X}$ and $\mathbf{Y}$ and the interaction among all labels $\mathbf{Y}$. When we have an energy function, the inference of the predicted label then simply becomes to minimize the energy function:

$$\hat{\mathbf{Y}}_U = \arg \min_{\hat{\mathbf{Y}}_U} E(\mathbf{X}, \mathbf{Y}_L, \hat{\mathbf{Y}}_U) \tag{4}$$

## 4.2 Training Cost-Augmented Inference Network

The training of SPENs usually takes the structured hinge loss (Tsochantaridis et al., 2004; Belanger & McCallum, 2016b) and relaxes the edge labels to be continuous for easier optimization. Assume the parameters for the energy function Eq. (3) is $\Theta$, our problem becomes:

$$\min_{\Theta} \max_{\hat{\mathbf{Y}}_L} [\triangle(\hat{\mathbf{Y}}_L, \mathbf{Y}_L)) - E_\Theta(\mathbf{X}, \hat{\mathbf{Y}}_L) + E_\Theta(\mathbf{X}, \mathbf{Y}_L)]_+ \tag{5}$$

where $\mathbf{Y}_L$ are the ground-truth DDI edges for training, and $\hat{\mathbf{Y}}_L$ are the predictions on these edges. $\triangle$ is the structured error function which returns a non-negative value to indicate the difference between the ground truth and prediction; and $[\cdot]_+$ means $\max(0, \cdot)$. In this paper, we use L1 loss as $\triangle$.

This above training loss is expensive to directly optimize due to the cost-augmented inference step. Following the idea of Tu & Gimpel (2018), we use a cost-augmented inference network (i.e. a graph neural network) $G_\Phi(\mathbf{X}, \mathbf{Y}_L)$ to approximate the output $\hat{\mathbf{Y}}_L$ in the training phase.

$$\min_{\Theta} \max_{\Phi} [\triangle(G_\Phi(\mathbf{X}, \mathbf{Y}_L), \mathbf{Y}_L) - E_\Theta(\mathbf{X}, G_\Phi(\mathbf{X}, \mathbf{Y}_L)) + E_\Theta(\mathbf{X}, \mathbf{Y}_L)]_+ \tag{6}$$

The problem can be seen as **minimax game** and optimized by alternatively optimizing $\Theta$ and $\Phi$.

**(1)** Fix $\Theta$, we can optimize the parameters for the cost-augmented inference networks

$$\max_{\Phi} [\triangle(G_\Phi(\mathbf{X}, \mathbf{Y}_L), \mathbf{Y}_L) - E_\Theta(\mathbf{X}, G_\Phi(\mathbf{X}, \mathbf{Y}_L)) + E_\Theta(\mathbf{X}, \mathbf{Y}_L)]_+ \tag{7}$$

**(2)** Fix $\Phi$, optimize $\Theta$:

$$\min_{\Theta} [\triangle(G_\Phi(\mathbf{X}, \mathbf{Y}_L), \mathbf{Y}_L) - E_\Theta(\mathbf{X}, G_\Phi(\mathbf{X}, \mathbf{Y}_L)) + E_\Theta(\mathbf{X}, \mathbf{Y}_L)]_+ \tag{8}$$

## 4.3 Semi-supervised Joint Training and Inference

The trained cost-augmented inference network cannot be directly used for test inference because of the cost augmentation. Notice that objective function to infer the unknown labels in the testing phase is Eq. (4). Due to the additional term in the loss function Eq. (8) during training, the derived training inference network cannot lead to a minimized energy. Thus we need another test inference network to optimize Eq. (4) in the testing phase.

We propose the test inference network $\hat{\mathbf{Y}}_U = G_\Psi(\mathbf{X}, \mathbf{Y}_L)$ to directly approximate the test output. Since the test edge features are not given(instead, we only know the indices of edges to test), so this test inference network also use $\mathbf{X}$ and $\mathbf{Y}_L$ as its inputs. In the DDI prediction problem, the training set and test set share the same set of nodes $\mathbf{X}$ in a single graph. In this graph, the energy optimization in the test phase should not be regarding only $\mathbf{Y}_U$ but the all graph including both $\mathbf{Y}_L$ and the predicted $\hat{\mathbf{Y}}_U$, as we showed in Eq. (4). Since this energy function is a graph neural network, that is to say, we use both $\mathbf{Y}_L$ and the prediction $\hat{\mathbf{Y}}_U$ as edge attributes for the computation of Eq. (4). When the parameter $\Theta$ for the energy function is given, the objective for this test inference network is as below:

$$\min_{\Psi} E_\Theta(\mathbf{X}, \mathbf{Y}_L, G_\Psi(\mathbf{X}, \mathbf{Y}_L)) \tag{9}$$

The previous methods (e.g. Tu & Gimpel (2018)) first fine-tune the trained inference network on training data again with respect to the original inference objective without cost augmentation $\arg\min_\Phi E(\mathbf{X}, G_\Phi(\mathbf{X}, \mathbf{Y}_L))$. But the real effect of this fine-tuning is actually doubted (Tu & Gimpel (2018)). Also considering that the definition of training-set energy and the test-set energy are in the same graph, and the test inference network should not deviate too much from the training cost-augmented inference network, we are proposing a new joint training strategy as follows.

We design the test inference network to share some bottom layers with the training cost-augmented inference networks. Then we do the optimization with the two following steps:

**(1)** Fix $\Theta$, we optimize the parameters for both the cost-augmented inference networks and the test inference network.

$$\max_{\Phi,\Psi}[\triangle(G_\Phi^L(\mathbf{X}, \mathbf{Y}_L), \mathbf{Y}_L) - E_\Theta(\mathbf{X}, G_\Phi^L(\mathbf{X}, \mathbf{Y}_L)) + E_\Theta(\mathbf{X}, \mathbf{Y}_L)]_+ - \lambda_1 E_\Theta(\mathbf{X}, \mathbf{Y}_L, G_\Psi^U(\mathbf{X}, \mathbf{Y}_L)) \tag{10}$$

To better discriminate the two inference networks, we use $G^L$ to indicates the predictions on training edges, and $G^U$ are the predictions on and missing edges. $\lambda_1$ is a hyperparameter which is often set as 1 in practice. Notice that $\Phi$ and $\Psi$ are not independent but share some parameters.

To better understand the benefit of the joint training, let us rewrite the parameter $\Phi$ and $\Psi$ into $\{\Phi_0, \Phi_1\}$ and $\{\Phi_0, \Psi_1\}$, where $\Phi_0$ is the shared parameter set between $\Phi$ and $\Psi$ for the shared layers of the two inference networks. Then the above optimization problem becomes:

$$\max_{\Phi_0,\Phi_1,\Psi_1}[\triangle(G_{\Phi_0,\Phi_1}^L(\mathbf{X}, \mathbf{Y}_L), \mathbf{Y}_L) - E_\Theta(\mathbf{X}, G_{\Phi_0,\Phi_1}^L(\mathbf{X}, \mathbf{Y}_L)) + E_\Theta(\mathbf{X}, \mathbf{Y}_L)]_+ \tag{11}$$

$$-\lambda_1 E_\Theta(\mathbf{X}, \mathbf{Y}_L, G_{\Phi_0,\Psi_1}^U(\mathbf{X}, \mathbf{Y}_L))$$

From this equation we can see the optimization of the second term, i.e. the test-set energy, can also impact the training of the cost-augmented inference network. If we fix all other parameters, the $\Psi_1$ still gives the test inference network enough flexibility to only optimize the test energy term; meanwhile the shared layers and the shared parameter $\Phi_0$ kept the two inference networks not deviate too much.

**(2)** Fix $\Phi$ and $\Psi$, we do not want the test inference network to impact the energy function (so that the objective for optimizing $\Theta$ is still from Eq. (6), so the min function is unchanged, i.e. still Eq. (8).

## 5 EXPERIMENT

We evaluate GENN[1] model to answer the following questions:

1. **Q1**: Does GENN provide more accurate DDI prediction than feed-forward GNNs? If so, we can demonstrate the usefulness of the new energy function.
2. **Q2**: Does GENN improves the supervised inference method (Section 4.2)? If so, we can demonstrate the effectiveness of our new optimization algorithm.
3. **Q3**: How does GENN respond to various fraction of edge missingness?
4. **Q4**: Does the model really capture meaningful label correlation?

### 5.1 EXPERIMENTAL SETUP

**Data** We implemented our experiments on two public datasets (with some modification of the setting): DeepDDI and Decagon. For both datasets, we randomly select 80% of the drug-drug pairs as training data, 10% as validation data and the remaining 10% as test data. For a given drug-drug pair, we predict its DDI types. Note that one drug-drug pair can have zero, one or multiple DDI types, so it is a multi-label prediction problem.

**DeepDDI Dataset** (Ryu et al., 2018) consists of 1861 drugs (nodes) and 222,127 drug-drug pairs (edges) from DrugBank which results in 113 different DDI types as labels. 99.87% drug-drug pairs only have one type of DDI. The edge exists when two drugs has at least one DDI type. The input feature for each drug is generated based on structural similarity profile (SSP), then its dimension

---

[1]https://github.com/ICLR2020-code/GENN.git

is reduced to 50 using principal components analysis (PCA) as suggested by DeepDDI. Moreover, Chemical-based similarity measure is used to reduce the the effect of redundant drugs on prediction accuracy where redundant similar drugs in training and test dataset will overestimate the predictive performanc as noted in Gottlieb et al. (2012). In particular, we use Tanimoto score (Tanimoto, 1957) and find 90% drug pairs has less than 0.4721 similarity score which show that there is no need to tackle the effect of redundant drugs.

**BIOSNAP-sub Dataset** (Marinka Zitnik & Leskovec, 2018) consists of 645 drugs (nodes) and 46,221 drug-drug pairs (edges) from TWOSIDES dataset which results in 200 different DDI types as labels. We extract data from  Zitnik et al. (2018) and only keep 200 medium commonly-occurring DDI types ranging from Top-600 to Top-800 that every ddi type has at least 90 drug combinations which contains proper number of edges for fast evaluation. 73.27% drug-drug pairs have more than one type of DDI. The input feature is 32-dimension vector by transforming one-hot encoding by Gaussian random projection using Pedregosa et al. (2011).

**Baselines** We compared GENN with the following baselines.

1. **Label Propragation (LP)** (Zhu et al., 2003) is a similarity-based semi-supervised method which makes use of unlabeled data to better generalize to new samples.
2. **MLP** is the model used in original DeepDDI dataset which accepts pairwise features (e.g. structural similarity profile (SSP)) between two drugs to predict DDI.
3. **DeepWalk** learns d-dimensional neural features for nodes based on a biased random walk procedure exploring network neighborhoods of nodes. For each drug pair, we concatenate the learned DeepWalk feature vectors and its original drug feature representation and use the MLP same as the above method to do classification.
4. **GNN** is a basic graph neural network based on MPNN framework as introduced in 3.2.
5. **GLENN** is designed to be an energy model with a CRF-style local energy function and the same inference algorithm as GENN. Notice that each node and all its connected edges form a clique (because these edges are mutual neighbors and they are also connected to each other due to the correlation), so we use $E = \sum_i f_1(\mathbf{x}_i + \sum_{j \in N(j)} f_2(e_{i,j}))$. The difference between Eq. (3) and Eq. (1) is that it does not include the neighbor node features in the equation. It is linear and has only one layer. We call it a graph local energy neural network (GLENN) and use the same optimization strategy as our final model.
6. **GENN$^-$** is an ablation model without semi-supervised joint training and inference.
7. **GENN** is our final model that incorporates the power of GNNs and energy models.

The **Implementation Details** can be found in Appendix B.

**Metrics.** To measure the prediction accuracy, we used the following metrics

1. **ROC-AUC**: Area under the receiver operating characteristic curve: the area under the plot of the true positive rate against the false positive rate at various thresholds.
2. **PR-AUC**: Area under the precision-recall curve: the area under the plot of precision versus recall curve at various thresholds.
3. **P@K**: short for **Precision at K** which is the mean percentage of correct predicted labels among TOP-K over all samples. Here, we specially choose P@1 and P@5 to validate performance as usually done in multi-label learning setting.

Following Decagon (Zitnik et al., 2018), we calculated the ROC-AUC and PR-AUC for each single label, and then use the average scores as the final values.

## 5.2 RESULTS

**Performance Comparison** For each dataset, we use 3 different random splits and run all the models on that split. The results are then averaged over 3 runs and we report the mean and std values in test dataset see Table 1. Note that the results on DeepDDI dataset in Table 1 is based on the 60% randomly sampled data, because it will take more than 2 days for models to converge with limited performance gain on total dataset.

From Table. 1, we can see DeepWalk and the methods based on MPNN framework outperform MLP and LP in both datasets with a large margin. It indicates the neighbor edges and nodes information

Table 1: Performance on DeepDDI and BIOSNAP-sub Dataset (LP caused memory error in Deep-DDI dataset).

|  | Method | P@1 | P@5 | PR-AUC | ROC-AUC |
|---|---|---|---|---|---|
| DeepDDI | LP | - | - | - | - |
|  | MLP | 0.7311 (.0026) | 0.1926 (.0005) | 0.5888 (.0362) | 0.9736 (.0032) |
|  | DeepWalk | 0.7773 (.0029) | 0.1962 (.0019) | 0.6276 (.0139) | 0.9786 (.0046) |
|  | GNN | 0.9002 (.0119) | 0.1986 (.0002) | 0.7606 (.0187) | 0.9861 (.0066) |
|  | GLENN | 0.8928 (.0067) | 0.1986 (.0002) | 0.7590 (.0095) | 0.9891 (.0058) |
|  | GENN⁻ | 0.9020 (.0220) | 0.1987 (.0004) | 0.8389 (.0512) | 0.9871 (.0071) |
|  | GENN | **0.9077 (.0293)** | **0.1990 (.0003)** | **0.8635 (.0286)** | **0.9928 (.0052)** |
| BIOSNAP-sub | LP | 0.1089 (.0049) | 0.0850 (.0040) | 0.0607 (.0013) | 0.6414 (.0010) |
|  | MLP | 0.2120 (.0019) | 0.1508 (.0009 ) | 0.1675 (.0022) | 0.8041 (.0009) |
|  | DeepWalk | 0.2463 (.0012) | 0.1719 (.0011) | 0.1908 (.0029) | 0.8311 (.0019) |
|  | GNN | 0.3275 (.0231) | 0.2215 (.0184 ) | 0.2494 (.0208) | 0.8757 (.0172) |
|  | GLENN | 0.3255 (.0192) | 0.2213 (.0181) | 0.2476 (.0209) | 0.8756 (.0172) |
|  | GENN⁻ | 0.3290 (.0102) | 0.2216 (.0137) | 0.2503 (.0175) | 0.8788 (.0159) |
|  | GENN | **0.3396 (.0072)** | **0.2326 (.0016)** | **0.2602 (.0034)** | **0.8855 (.0026)** |

actually help the representation learning which improve the performance. Compared with GNN, GLENN achieves nearly the same performance and gets little better score with respect to ROC-AUC. As for GENN, it outperforms others with respect to all metrics in both datasets which give the positive answer to **Q1** that GENN does provide more accurate DDI prediction. Especially, compared to GNN and GLENN, it demonstrates the power of our new designed graph neural network based global energy function. It is more expressive and flexible to capture the label correlation. In addition, for variants of GENN, the supervised learning model GENN⁻ setting performs worse compared with GENN which give the answer of **Q2**.

**Analysis of Robustness**

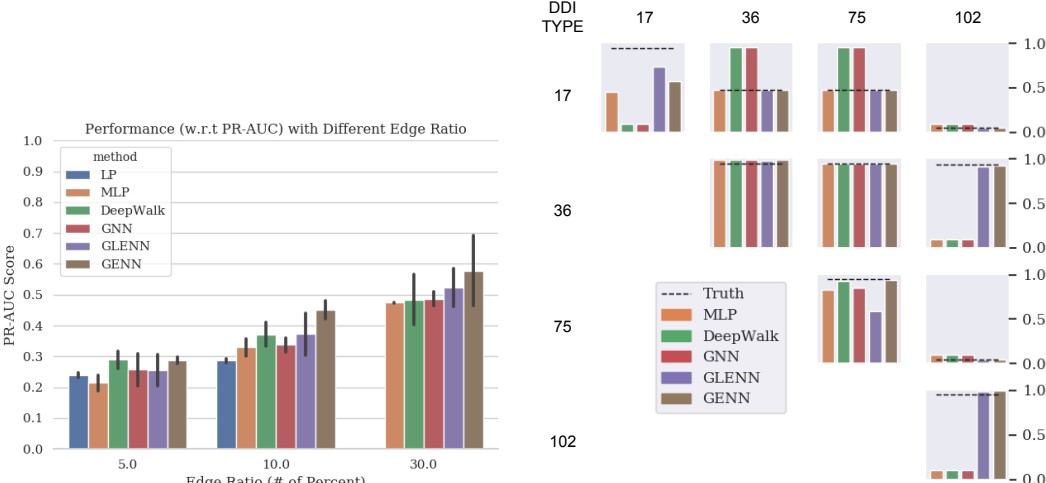

Figure 3: **Analysis of Robustness**. Methods are trained on 5%, 10% and 30% of edges on Deep-DDI dataset. Mean PR-AUC with standard deviation values are reported on the rest of edges. (Note that the performance of LP is missing at the ratio of 30% because of memory error.

Figure 4: **Correlation Analysis**. Pearson correlation coefficients between four randomly chosen DDI types. Each diagnoal subplot shows the correlations between ground truth and different methods, while upper diagonal subplot shows the pairwise correlations between DDI types.

We design experiments to show that compared with baseline models, GENN achieved empirically more robust performance in the setting of high edge missingness. We let each method be trained on 5%, 10%, and 30% of edges on DeepDDI dataset, and predict on the rest of them. From Figure. 3, we can see that GENN is better than all the baselines in every percentage point (**Q3**).

**Correlation Analysis**

In correlation analysis experiment, we chose the models trained on 60% DeepDDI dataset same as the models reported in Table. 1 and made DDI type predictions on the rest of the edges (40% left test dataset). To illustrate that GENN is able to capture correlation between labels, i.e., different DDI types by incorperating energy function, we further calculate the pearson correlation coefficent between *distribution* of DDI types. For each of the 113 DDI types[2], we define its *distribution* by counting the number of occurrences in the prediction on each drug node over all edges in test dataset.

The PairGrid figure 4 shows the pairwise relation between these randomly chosed four DDI types, numbered as 17, 36, 75, 102. For instance, DDI type 17 means "Drug *a* may increase the cardiotoxic activities of Drug *b*" and DDI type 36 means "Drug *a* may decrease the bronchodilatory activities of Drug *b*". For the ground truth, DDI type 17 has medium correlation with DDI type 36 but MLP and DeepWalk overestimate the coefficient. Especially for DDI type 102, only GLENN and GENN capture the true *distribution* same with the ground truth. On the one hand, it is consistent with the high performance achieved by GENN. On the other hand, the consistency between Truth and GENN demonstrates that GENN really capture some label correlation which answers **Q4**.

## 6 CONCLUSION

In this paper, we proposed GENN to cast DDI detection as a structure prediction problem with a new GNN based energy function. Experiments on two real DDI datasets demonstrated that GENN is superior to the models without consideration of link type correlations and achieved up to 0.8635 (13.53% relative improvement) with respect to PR-AUC. Future works includes extending the correlated GNNs to heterogeneous networks and incorporating medical domain knowledge with structure information such as drug classification ontology in the learning.

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

# A  NOTATIONS

Table 2: Notations used in GENN.

| Notation | Definition |
|---|---|
| $G : \{\mathcal{V}, \mathcal{E}\}$ | the whole graph data with nodes, edges set $\mathcal{V}, \mathcal{E}$ |
| $\mathbf{X} \in \mathbb{R}^{N \times D}$ | node feature matrix with $N$ nodes and features dimension $D$ |
| $\mathbf{e}_{i,j} \in \{0, 1\}^L$ | DDI type vector between nodes $i, j$ with $L$ different types |
| $\mathbf{Y} \in 0, 1^{|\mathcal{E}| \times L}, \hat{\mathbf{Y}} \in [0, 1]^{|\mathcal{E}| \times L}$ | ground truth labels and prediction labels of all graph |
| $\mathbf{h}_i^t, \mathbf{h}_i \in \mathbb{R}^M$ | $i$-th node's feature at $t$-th update and final out of MPNN |
| $G(\cdot)$ | graph neural networks (GNN) |
| $\mathbf{B}, \mathbf{U}, \mathbf{W}_*, \mathbf{b}$ | learnable weight matrices and bias |
| $E_\Theta(\mathbf{X}, \mathbf{Y})$ | energy function parameterized by $\Theta$ |
| $G_\Phi(\mathbf{X}, \mathbf{Y}), G_\Psi(\mathbf{X}, \mathbf{Y})$ | training and testing inference network |

# B  IMPLEMENTATION DETAILS

Most methods are implemented in PyTorch-v1.1.0 (Paszke et al., 2017) based on Fey & Lenssen (2019) and trained on an Ubuntu 18.04 with 56 CPUs and 64GB memory.

We use the implementation from Pedregosa et al. (2011) for Label Propagation method with the setting of 0.25 gamma value for rbf kernel and 200 maximum number of iteration. As for DeepWalk method, the official implementation[3] is used with the setting of embedding dimension as 50 and the length of random walk as 20.

We implement training inference network ($G_\Phi$), test inference network ($G_\Psi$) and energy network $E_\Theta$ with the same 2-layer message passing neural network (MPNN) encoder and 1-hidden-layer MLP decoder structure. Hidden dimension are all set as 100, and to reduce the number of parameters and ensure the two inference networks ($G_\Phi$ and $G_\Psi$) do not deviate too much, we share the message passing layer for them, while make the MLP layer different. For energy network $E_\Theta$, we also use message passing for node encoding, but the layer has a different set of parameters from the inference networks in order to make the minimax training schema more effective. For MLP decoder, we use batch-normalization technology (Ioffe & Szegedy, 2015) with Relu non-linear activation function.

For training efficiency, we first train the basic MPNN (i.e. GNN baseline) on the training data, and then initialize all MPNN layers used in inference networks and energy networks with the these pre-trained parameters. We use the learning rate 0.01 for methods to train and 0.001 to fine tuning on two datasets with early-stop mechanism (i.e., training is stopped when there is no performance improvement on validation dataset with 35 consecutive epochs). Threshold for prediction is simply set as 0.4.

# C  CROSS ENTROPY REGULARIZATION

From the experience of Tu & Gimpel (2018), adding an local cross entropy loss to the energy function could improve the performance a lot. It can be seen as a multi-task training with two objectives: energy optimization and reconstruction error minimization. We add the reconstruction cross-entropy loss of both the training and test inference networks in Eq. (10) to train better inference network approximations.

$$\max_{\Phi, \Psi}[\triangle(G_\Phi^U(\mathbf{X}, \mathbf{Y}_L), \mathbf{Y}_L) - E_\Theta(\mathbf{X}, G_\Phi^U(\mathbf{X}, \mathbf{Y}_L)) + E_\Theta(\mathbf{X}, \mathbf{Y}_L)]_+ \qquad (12)$$

$$-\lambda_1 E_\Theta(\mathbf{X}, G_\Psi^{all}(\mathbf{X}, \mathbf{Y}_L)) - \lambda_2 L_\Phi^{CE} - \lambda_3 L_\Psi^{CE}$$

where $L_\Phi^{CE} = \mathcal{L}_{CE}(G_\Phi^U(\mathbf{X}, \mathbf{Y}_L), \mathbf{Y}_L)$, $L_\Psi^{CE} = \mathcal{L}_{CE}(G_\Psi^U(\mathbf{X}, \mathbf{Y}_L), \mathbf{Y}_L)$ and $\lambda_2, \lambda_3$ are hyperparameters which can all be set as 1 the same as $\lambda_1$ in practice. Note that these additional regularization terms are independent of $\Theta$, so we do not need to add it when minimizing over $\Theta$ (Eq. (8) is still unchanged).

---

[3]https://github.com/phanein/deepwalk

Table 3: Correlation Analysis Between Pair of DDI Type. (↑ means increase while ↓ means decrease).

| Pair of DDI Types | | Truth | GENN | GNN |
|---|---|---|---|---|
| Metabolism ↓ | Bradycardic Activities ↑ | 0.6423 | 0.5375 | 0.3176 |
| Risk of Hypotension ↑ | Neuromuscular Blocking Activities ↓ | 0.3347 | 0.2954 | -0.0027 |
| Risk of Hyperkalemia ↑ | Neuromuscular Blocking Activities ↑ | 0.2854 | 0.2822 | 0.3913 |

## D    CORRELATION ANALYSIS

As shown in table 3, there are three pairs of DDI type which shows large (0.6423), medium (0.3347) and small postive correlation (0.2854) indicated by Truth column. For the first pair of DDI type, decreasing metabolism has large correlation with increasing the bradycardic activities in ground truth but GNN only got medium correlation coefficient much lower than GENN'. As for the last two pairs of DDI type, GNN performed bad and gave a nearly zero coefficient for the second and overestimated the coefficient for the last. In the meantime, GENN corresponds with the ground truth which in some extent gives the evidence that GENN has the power to capture correlation between labels.

