# OpenReview forum: "GENN: Predicting Correlated Drug-drug Interactions with Graph Energy Neural Networks"
_ICLR.cc/2020/Conference — Reject_

### Official Review · AnonReviewer2 · 2019-10-22
**Official Blind Review #2**

**Rating:** 6

**Review:**

Abstract:
Understanding drug-drug interactions (DDI) is an important task in drug development and prescription management. The authors proposed a new graph energy neural network (GENN) for DDI prediction. Comparing to the existing Decagon model (Zitnik et al. 2018), the proposed new model considered correlations between DDI types and used a new energy function to capture this information. By comparing to the previous baselines, the authors demonstrated their approach was able to achieve the SOAT performance in terms of prediction accuracy, be more robust to missing DDI data, and better capture the correlations between DDI types.

Major comments:
The authors succeed in combining graph neural networks and structure prediction to capture correlations of DDI types between drug-drug pairs.   They used an energy function based on graph neural networks to capture the dependencies among edge types in DDI graphs.

However, the description of training cost-augmented inference network and test inference network is convoluted. The authors didn't elaborate on why trained cost-augmented inference network cannot be directly used for test inference, and why the prediction for unknown DDI type is different in test inference.  In addition, the notations used in the last sentence of the second paragraph of section 4.3 seem to be inconsistent with Eq. (4)

The authors compared their model (GENN) with multiple baselines and demonstrated superior performance only in PR-AUC. In the description for Table 1, the authors claimed: "GENN largely outperforms others with respect to all metrics in both datasets."  However, this conclusion can not be supported by Table 1, in which GENN only shows trivial improvement comparing to GNN for the most reported metrics.

Last but not least, the authors did not provide any limitations of their work.

Minor comments:
In general, the paper is well written and easy to follow, except for sections 4.2 and 4.3.  As mentioned above, the notations used in the last sentence of the second paragraph of section 4.3 seem to be inconsistent with Eq. (4)

**Experience Assessment:**

I have read many papers in this area.

**Review Assessment: Checking Correctness Of Derivations And Theory:**

I did not assess the derivations or theory.

**Review Assessment: Checking Correctness Of Experiments:**

I assessed the sensibility of the experiments.

**Review Assessment: Thoroughness In Paper Reading:**

I made a quick assessment of this paper.

---

> ### Author Response · Authors · 2019-11-15
> **Reply to Reviewer#2**
>
> Thank you for your detailed comments. We corrected the typo in the second paragraph of section 4.3; and changed the expression “largely outperforms” by removing the word “largely”. Despite some confusion w.r.t. P@1 and P@5, for the other metrics (PR-AUC and ROC-AUC), actually our final model GENN did outperform others a lot. The other questions are answered as follows. And please also refer to our revised paper (we listed the updates in the above common comment).
>
> Q1:The authors didn't elaborate on why trained cost-augmented inference network cannot be directly used for test inference, and why the prediction for unknown DDI type is different in test inference.
> A1: Thank you for proposing this question. We updated the paper and added more explanations in Section 4.2 to make this point more clear. It is actually a known conclusion from previous works for structure prediction. The cost-augmented inference network in the training phase has a different optimization goal from the test phase: the objective for training is Eq. (5) while the test objective is Eq. (4). If we use the training inference network to get the optimized $Y$, it will not get the minimized energy, due to an additional term in the objective function $\delta(Y, \hat{Y})$. That is why it is called cost-augmented inference during training.
>
> Q2:The authors did not provide any limitations of their work
> A4: We have mentioned the limitations of our method in the conclusion section that the current version of GENN only work on the graph with the same type of nodes. It leaves a future work to extend it to heterogeneous networks where nodes have different types such as protein type and drug type.

---

### Official Review · AnonReviewer3 · 2019-10-24
**Official Blind Review #3**

**Rating:** 3

**Review:**

This paper proposes a framework to learn correlated drug-drug interaction based on structured prediction energy networks (SPEN). The core idea is to model the dependency structure of the labels (multi-label) by minimizing a designed energy function. The graph energy is designed as MLP over the mean of all nodes embeddings, where the nodes embeddings are obtained through a graph convolutional network. The edge information is included in the node embedding when aggregating neighborhood information. The proposed method also introduces an additional test inference network to jointly train with the cost-augmented training network under the semi-supervised setting. The authors tested on two DDI datasets and the result shows improvement compared to several baseline methods.

Strengths
- motivation: the authors consider the correlations between DDI labels (multi-label) which could potentially improve prediction of DDI. The proposed method uses structure prediction energy networks to model such dependency.

- method: the proposed method introduce an additional test inference network to fit the energy minimization framework into a semi-supervised setting.

Weakness
- The core part of the proposed method which differs from previous works needs to be elaborated more.
- The performance improvement seems to be marginal, especially on the second dataset.

Detailed comments
- The work is based on structured prediction energy networks (SPEN, Belanger, et al. 2016) and its follow-up works (eg., Belanger et al. 2016, Lifu Tu et al. 2018), the overall framework is very similar with previous work, where the feature extraction part is replaced by a graph convolutional network module. The core difference lies in the additional testing inference network which is jointly trained for adapting supervised SPEN into a semi-supervised setting. This part actually differs from all previous works and needs to be elaborated more. Why does such design make sense and what benefit can be brought through such design? The formulation for semi-supervised SPEN could be defined more clearly and worth elaboration.

- The experiments are run on 3 different random splits, based on the mean(std) of the evaluation metric, the performance of the proposed method does not vary much compared to baselines, especially on the second dataset, eg., GNN 0.25 +/- 0.02 compared with GENN 0.26. Also, GLENN < GNN seems to imply including energy is not the most important part for helping the task, but rather the semi-supervised joint training truly improves the performance.

**Experience Assessment:**

I have read many papers in this area.

**Review Assessment: Checking Correctness Of Derivations And Theory:**

N/A

**Review Assessment: Checking Correctness Of Experiments:**

I carefully checked the experiments.

**Review Assessment: Thoroughness In Paper Reading:**

I read the paper thoroughly.

---

> ### Author Response · Authors · 2019-11-15
> **Reply to Reviewer#3**
>
> Thank you for your comments.
> Q1:The core part of the proposed method which differs from previous works needs to be elaborated more.
> A1: As you suggested, we modified our paper and explain more clearly about the difference from previous methods in the Introduction. Please also refer to the related work where we explained the difference between the most related works. To summarize, first we need to clarify that the paper is not focusing on developing a new inference method to improve general SPENs. Instead, we are targeting on combining the ideas of energy models with graph neural networks (GNNs) to improve the GNNs. Compared to previous models using energies in graph neural networks, we have a new energy definition from SPENs, which can deal with more flexible, more global correlation; compared with previous models which focus on the inference of general SPENs, we improve the inference algorithm by jointly training the cost-augmented inference network and test inference network to adapt our special task in the context of graph learning.
>
>
> Q2: Why does “adapting supervised SPEN into a semi-supervised setting” make sense and what benefit can be brought through such design?
> A2: Thank you for pointing out this. We updated the Section 4.3 and elaborated more on this point. In addition to the revised paper, let us briefly explain it there. Graph learning is generally a semi-supervised setting, as shown in GCN [1]. The node embeddings are usually obtained from both training and test nodes(or edge). The defined graph energy is derived from node embeddings, so using all the graph to get the energy is one reason for semi-supervised learning. In addition, we shared some parameters between training and testing inference networks, jointly optimizing the training and testing objectives could lead to better results as we explained in the revised paper. That is another reason. The benefit is also proved in the experiments by comparing with GENN-.
>
> [1] Kipf T N, Welling M. Semi-Supervised Classification with Graph Convolutional Networks. ICLR 2016.
>
> Q3:experimental performance
>
> A3:GLENN is a variant of GNN using local energy and the same semi-supervised joint optimization algorithm, so “GLENN<GNN” indicates the global energy (our GNN based energy function) is better than local energies. Since all baselines are tested in semi-supervised setting and GENN performs better on both datasets,  we can safely attribute the performance gain to the effectiveness of the energy design, instead of the usefulness of semi-supervised learning. The semi-supervised learning is useful by comparing GENN and GENN-.
>
> As to the performance gain, we believe for most models the performance is data dependent. 1% increase of PR-AUC is not marginal for a difficult task. As this work is highly related to graph neural networks, let us elaborate some recent research in this community. From GCN to every new developed GNN model (e.g. GraphSAGE, Graph Infomax, GAT, GMNN), the increased accuracy is usually just around 1% on general benchmark datasets (Cora, Pubmed, Citeer).

---

### Official Review · AnonReviewer1 · 2019-10-26
**Official Blind Review #1**

**Rating:** 3

**Review:**

This paper presents a graph neural network for drug-to-drug interaction (DDI) prediction, which explicitly models link type correlation. Basically, the drug-to-drug interaction prediction problem is a specific type of link prediction task, with drugs as vertices and interaction as edges, and the authors propose a graph neural network with an energy-based formulation where the link types are encoded as the graph edges. The authors validate their method against feedforward GNNs on two DDI prediction datasets, and achieve significantly improved performances.

Pros
- The drug-to-drug interaction prediction is a relatively less explored application of deep learning with growing interests.
- The proposed energy-based formulation that considers link type correlation intuitively makes sense and performs significantly better than some of the existing GNN approaches.

Cons
- Using energy-based formulation for graph neural network is not novel, and thus the paper lacks novelty in methodology point of view. Thus this paper seems to fit better for a pharmaceutical journal or a journal rather than a top-tier deep learning conference such as ICLR. I do not think the paper will be of interest to a large audience.

- The experimental validation is limited. The authors only compare against GNN and GLENN, without even mentioning other relevant baselines such as GCN and GAT. Thus I suggest the authors to perform a more extensive comparative study.

- The qualitative analysis is insufficient, even as a domain-specific empirical study paper. It would be better if the authors included actual examples of drug-drug interaction predicted by the proposed model and an existing model, for further analysis and interpretation.

In sum, while I believe that DDI prediction is an interesting application of deep learning with growing interests, I do not believe the paper has sufficient novelty or experimental validation / qualitative examples to be a meaningful contribution to ICLR. I suggest the authors to work on the experimental validation and qualitative analysis part and submit it to a workshop or a pharmaceutical journal instead.


**Experience Assessment:**

I have read many papers in this area.

**Review Assessment: Checking Correctness Of Derivations And Theory:**

I carefully checked the derivations and theory.

**Review Assessment: Checking Correctness Of Experiments:**

I carefully checked the experiments.

**Review Assessment: Thoroughness In Paper Reading:**

I read the paper thoroughly.

---

> ### Author Response · Authors · 2019-11-15
> **Reply to reviewer#1**
>
> Thank you for the comments. Although wrapped with an application to the drug-drug-interaction prediction tasks, this paper is essentially not a domain-specific empirical study paper but rather a new graph learning method for link prediction. Its model is new compared to all previous methods, and it does not require much knowledge about medical domain.
>
> Q1:Using energy-based formulation for graph neural network is not novel, and thus the paper lacks novelty in methodology point of view.
> A1: We admit this is not the first work to combine GNN and energy models, but how to formulate the energy, how to do inference, and how to apply to a real-world task is not trivial.
> We have clearly described the difference between our paper with the previous energy-based GNN models in Section 2.2. And we also add more explanation in the Introduction section of the revised paper. To summarize, the differences include: 1. There is no existing work that defines the energy function as a graph neural network. Instead, most of them are based on the CRF-type energies, which captures only local correlations between neighbor nodes (whose disadvantage was demonstrated in our experiments with GLENN). And no previous one can deal with the problem of multi-typed link (e.g. DDI) prediction. 2. We also introduce a new inference method that is customized in the context of graph learning with our new energy function. 3. We believe given the new energy formulation and the new inference method, our model is novel. We hope the reviewer can take a second look at it and reevaluate the novelty.
>
> Q2:This paper seems to fit better for a pharmaceutical journal or a journal rather than a top-tier deep learning conference such as ICLR. I do not think the paper will be of interest to a large audience.
> A2: As we formulate in the paper, DDI prediction is a typical multi-type link prediction problem, which is one of the classical tasks for graph learning. In our paper we in fact provided a new general model for multi-type link prediction tasks instead of a DDI-specific solution. We chose DDI because its “label correlation” is easily understood as well as the task can be evaluated based on large real-world data. It is not fair to say it does not attract larger audience. Here we can elaborate two classical GNN models as examples: “Convolutional Networks on Graphs for Learning Molecular Fingerprints (Duvenaud et al. 2015)”, “Neural message passing for quantum chemistry (Glimer et al. 2017)”. We believe no one will regard them as chemical/biological papers even though their titles are about molecular fingerprints and quantum chemistry.
> Similarly, the authors of this paper are actually not from pharmaceutical domain, and the main contribution of the paper is to provide a new method in the graph learning community.
>
>
> Q3:The experimental validation is limited. The authors only compare against GNN and GLENN, without even mentioning other relevant baselines such as GCN and GAT.
> A3: GCN and GAT are obviously not suitable for the multi-type link prediction tasks. The original GCN and GAT models do not take edge type information into consideration. We do not see any reason to mention them. In contrast, we have used one of the state-of-the-art message passing neural networks (Glimer et al. 2017) (which integrates edge type information in its message passing schema) as our base model, which we believe is representative enough to demonstrate the usefulness of the energy modules. In addition we also developed GLENN and GENN- for ablation study, which helped demonstrate the effectiveness of our new modules.
>
> [Duvenaud et al. 2015] Convolutional Networks on Graphsfor Learning Molecular Fingerprints. In NIPS2015.
> [Glimer et al. 2017] Neural Message Passing for Quantum Chemistry. In NIPS2017.

---

### Author Response · Authors · 2019-11-15
**Summary of updates in the revised paper**

1. We added more elaboration about the difference from previous works in the contributions of Introduction. We also add one sentence in the second paragraph of Section 2.2 to explain our difference from the previous method.
2. We re-organized the paragraphs in Section 4.3 and also added more detailed formulations and explanations for semi-supervised joint training.
3. We corrected the typo $Y_L$ and $\hat{Y}_U$ in the second paragraph of Section 4.3.
4. We added an explanation in the first paragraph of Section 4.3 to explain why cost-augmented inference network cannot be used in the testing phase.
5. We added two sentences after the first two questions of the experimental section to further explain the meaning of the comparison.

---

### Decision · Program_Chairs · 2019-12-19

**Decision:**

Reject

**Comment:**

This paper studies the use of a graph neural network for drug-to-drug interaction (DDI) prediction task (an instance of a link prediction task with drugs as vertices and interaction as edges). In particular, the authors apply structured prediction energy networks (SPEN) and model the dependency structure of the labels by minimising an energy function. The authors empirically validate the proposed approach against feedforward GNNs on two DDI prediction tasks. The reviewers feel that understanding drug-drug interactions is an important task and that the work is well motivated. However, the reviewers argued that the proposed methodology is not novel enough to merit publication at ICLR and that some conclusions are not supported by the empirical analysis. For the former, the benefits of the semi-supervised design need to be clearly and concisely presented. For the latter, providing a more convincing practical benefit would greatly improve the manuscript. As such, I will recommend the rejection of this paper at the current state.